# Research on Werner Syndrome: Trends from Past to Present and Future Prospects

**DOI:** 10.3390/genes13101802

**Published:** 2022-10-06

**Authors:** Kyoshiro Tsuge, Akira Shimamoto

**Affiliations:** Department of Regenerative Medicine Research, Faculty of Pharmaceutical Sciences, Sanyo-Onoda City University, 1-1-1 Daigaku-Street, Sanyo-Onoda Yamaguchi 756-0884, Japan

**Keywords:** premature aging, RecQ helicases, chromosomal instability, senescence, telomere, DNA repair, ribosomal DNA, induced pluripotent stem cells, tissue stem cells, disease models

## Abstract

A rare and autosomal recessive premature aging disorder, Werner syndrome (WS) is characterized by the early onset of aging-associated diseases, including shortening stature, alopecia, bilateral cataracts, skin ulcers, diabetes, osteoporosis, arteriosclerosis, and chromosomal instability, as well as cancer predisposition. *WRN*, the gene responsible for WS, encodes DNA helicase with a 3′ to 5′ exonuclease activity, and numerous studies have revealed that WRN helicase is involved in the maintenance of chromosome stability through actions in DNA, e.g., DNA replication, repair, recombination, and epigenetic regulation via interaction with DNA repair factors, telomere-binding proteins, histone modification enzymes, and other DNA metabolic factors. However, although these efforts have elucidated the cellular functions of the helicase in cell lines, they have not been linked to the treatment of the disease. Life expectancy has improved for WS patients over the past three decades, and it is hoped that a fundamental treatment for the disease will be developed. Disease-specific induced pluripotent stem (iPS) cells have been established, and these are expected to be used in drug discovery and regenerative medicine for WS patients. In this article, we review trends in research to date and present some perspectives on WS research with regard to the application of pluripotent stem cells. Furthermore, the elucidation of disease mechanisms and drug discovery utilizing the vast amount of scientific data accumulated to date will be discussed.

## 1. Introduction

Twenty-six years ago, *WRN*, the gene responsible for the premature aging associated with Werner syndrome (WS), was identified [1]. Research on genes that suppress aging has been conducted worldwide. During this time, many researchers have participated in these investigations, and it has been established that the protein encoded by *WRN* is an ATPase activated by single-stranded DNA with DNA unwinding activity in the 3′ to 5′ direction [2], and that it performs a unique 3′ to 5′ exonuclease activity not observed in other RecQ family members [2], interacts with a great variety of DNA metabolic proteins, and that it is involved in replication, repair, recombination, transcription, and histone modifications to maintain chromosome stability from the base sequence level to the chromatin level [3]. Within the enzymes/proteins that play crucial roles in these chromosomal events, the WRN helicase plays a fine-tuning role as a supporter. Its functional abnormalities induce chromosomal instability. Abnormal DNA structures accumulated in chromosomes and changes in gene expression profiles caused by epigenetic and transcriptional abnormalities lead to systemic disruption of cellular functions (e.g., loss of protein homeostasis, mitochondrial dysfunction, shortened mitotic lifespan, impaired differentiation) and manifest as symptoms such as premature aging [3].

Because studies using cultured cancer cells and normal fibroblasts have limitations in examining the mechanisms of pathogenesis, iPS cells derived from WS patients are used to recapitulate pathological conditions and are applied in regenerative medicine following induction of differentiation into various tissue cells [4,5]. In addition, the iPS cells with restored mutant *WRN* genes were established by genome editing for a rigorous analysis of the mechanisms [6].

There are five RecQ homologs in the mammalian genome. *RecQL1* is the *E. coli* RecQ ortholog, and it is responsible for RECON syndrome [7]. *RecQL2* and *RecQL3* are the *BLM* and *WRN* genes responsible for Bloom and Werner syndromes, respectively [2]. The discovery of the *WRN* gene has highlighted the human RecQ homologs, *RecQL4* [8] and *RecQL5* [8,9], and has led to the identification of Rothmund-Thomson syndrome as a RecQ disease [10], leading to the idea of an association between chromosomal instability and premature aging. In this article, we introduce recent trends in the study of WRN helicases and RecQ homologs and discuss future possibilities.

## 2. Role in Replication Stress Response and DNA Repair

WRN helicases localize to the nucleolus during steady-state cell cycle progression, but translocate to the site of DNA damage when it happens. WRN helicases are presumed to play a critical role in replication fork reassembly because they co-localize with RPA and Rad51 and form a focus at damaged and stalled replication forks [11]. Recent identifications of various interaction factors have helped to illustrate their role in cellular activities. At the reversal site of the nascent DNA strand generated by the arrested replication fork, WRN is thought to support replication fork reassembly through its exonuclease activity, or that of a bound DNA2 helicase [12,13]. On the other hand, at the double-strand break site of the DNA generated by the arrested replication fork, WRN binds to the nascent DNA strand with NBS1, regulating the nuclease activity of MRE11 to stabilize Rad51, and supporting replication fork reconstruction by preventing the nascent DNA strand from degrading [14]. In both cases, if the WRN does not function, small-scale DNA strand degradation occurs around the replication fork caused by other DNA repair pathways, resulting in the deletion of genetic information and abnormal chromosome structure (Figure 1a).

It is well known that WRN binds to Ku, a non-homologous end-joining (NHEJ) factor, and repairs DNA double-strand breaks other than replication forks in the NHEJ pathway [2]. The Ku heterodimer binds to the DNA ends of double-strand breaks, and DNA repair occurs through the binding of non-homologous ends. The MRN complex (MRE11/RAD50/NBS1) is also known to bind to the DNA ends of double-strand breaks and repair them by non-homologous end joining with end resection [2]. In the process by which either the Ku heterodimer or the MRN complex are selected to bind to the DNA ends of double-strand breaks, the WRN binds to the DNA ends of the breaks together with the Ku dimer, inhibits the recruitment of the MRN complex, promotes non-homologous end-joining repair, and prevents repair involving deletion of genetic information due to end resection [15] (Figure 1b).

The RNA polymerase complex, which synthesizes mRNA from chromosomal DNA, is temporarily stalled during the mRNA synthesis when it encounters DNA damage, such as DNA double-strand breaks. Marabitti et al. report that, compared with normal cells, the R-loop accumulates more in the fibroblast cells from WS patients under replication stress [16]. The R-loop is formed when transcription complexes collide with replication forks that have been arrested under replication stress. Since the R-loop typically degrades promptly in normal cells, and transcription and replication resume [17], WRN may play a role in the R-loop degradation. On the other hand, the degradation of the R-loop, which accumulates upon collision of Pol II complexes with replication forks in WS patient cells, is mediated by the XPG-mediated pathway responsible for the degradation of the R-loop formed by chromosomal DNA double-strand breaks [16,18]. These studies suggest that the cells preferentially utilize the WRN function when a specific type of DNA damage occurs. In the cells of patients with a *WRN* deficiency, another pathway responds and repairs the DNA damage possibly associated with the WRN, accompanied by the deletion of genetic information and resulting abnormal chromosome structure, which in turn results in premature mitotic senescence (Figure 1a).

## 3. Role in the Ribosomal DNA Regions

The ribosomal RNA gene (rDNA) locus in the eukaryotic cells consists of more than 100 copies of the same transcription unit, and such repetitive sequences are prone to recombination between the copies, which may contribute to chromosomal instability [19]. In budding yeast, it has been shown that SIR2 suppresses undesirable recombination, and that FOB1 increases the copies to maintain an appropriate rDNA copy number and suppress cellular senescence [19]. *Sgs1* is an ortholog of the RecQ helicase with which *BLM* and *WRN* are functionally complementary to the gene [20]. In a mutant strain of *Sgs1*, an accumulation of arrested replication forks and DNA double-strand breaks at the ribosomal DNA replication fork barrier (RFB) was observed. Additionally, the SGS1 helicase functions in concert with DNA2 helicase and FOB1 to maintain the rDNA locus [21].

WRN proteins have been reported to localize to the nucleolus in many cell types [22,23], and to migrate from the nucleolus to the nucleoplasm following certain types of DNA damage [11,24]. The reported role of WRN in the nucleolus is to promote transcription of ribosomal RNA by forming a complex with RNA polymerase I [25]. However, to date, little is understood about the involvement of WRN helicase in rDNA maintenance in mammalian cells because their highly repetitive nature has hindered sequence-level analysis. We recently used Oxford Nanopore sequencing technology to show that the human rDNA array maintains a highly regular and uniform rDNA repeat structure [26]. On the other hand, the rDNA array of WS patient cells showed an increased frequency of mutated copies, suggesting that this may be one of the causes of premature senescence [26] (Figure 1c). In addition, the frequency of mutations was lower in the rDNA array of the WS iPS cells compared with the original patient fibroblast cells, suggesting that cells with a stable rDNA array may have been selected during the reprogramming process [26].

## 4. Role in Telomere Maintenance

The telomeres at the end of a chromosome are unique structures consisting of the 5′-(TTAGGG)n-3′ sequence repeated more than 2500 times, with specific proteins binding to the repeat sequences and protecting chromosome ends from the DNA damage response. Telomeres also play an important role in cancer suppression because they shorten during cell division in normal cells due to the terminal replication problem. It is known that the mitotic life span of the fibroblasts from WS patients is much shorter than in those of healthy individuals [27]. Since abnormalities of telomeres have been reported in the cells derived from WS patients [28,29], the premature senescence observed in the cultured fibroblasts cells from WS patients may be due to abnormal telomere functions. The introduction of telomerase, a telomere-extending enzyme, into fibroblast cells from WS patients extends telomere length and prevents the premature senescence phenotype [30]. Reprogramming of the cells by introducing Yamanaka 4 factors (i.e., Oct3/4, Sox2, Klf4, and c-myc) activates the endogenous *hTERT* gene and immortalizes patient cells [4,5]. These results indicate that WRN helicase plays an important role in maintaining telomere structure.

Analysis using recombinant proteins has revealed that WRN helicase is an ATPase activated by single-stranded DNA with DNA unwinding activity in the 3′ to 5′ direction [31,32,33], and that it performs a unique 3′ to 5′ exonuclease activity not observed in other RecQ family members [34]. WRN helicase also performs structure-specific DNA unwinding activity and is suitable for solving G-quadruplex and recombination intermediates of Holliday structures [35,36,37]. Crabbe et al. showed that WRN helicase is required for the nascent lagging strand synthesis of telomeres [38]. Of the template strands that are unwound by the replicative helicase, the nascent lagging strand is synthesized by replicative DNA polymerase using telomere 5′-(TTAGGG)n-3′ as a template. Single-stranded G-rich telomeric DNA (i.e., TTAGGG) forms a G-quadruplex structure. It has been suggested that WRN helicase supports the synthesis of the nascent lagging strand by replicative DNA polymerase while resolving the quadruplex structure [38]. In support of this, WRN helicase has been observed to stimulate DNA polymerase δ to proceed with DNA synthesis by overcoming G-rich repeats, and to promote the maturation of the Okazaki fragment in vitro [39] (Figure 1d).

On the other hand, chromosome ends are sealed by telomere-specific DNA-protein complexes, in which telomere-binding proteins, such as shelterin (TRF1, TRF2, RAP1, TIN2, TPP1, and POT1), bind to telomere sequences [40]. Among these shelterin components, WRN helicase interacts with TRF1, TRF2, and POT1, and has been shown to play a critical role in telomere homeostasis [41,42,43].

Lombard et al. report that, in vivo, mice with C-terminally truncated WRN protein without a helicase domain were viable for more than 2 years and did not show signs of premature aging [44]. Chang et al. also examined the effect of *WRN* knockout on the null background of Terc (a telomerase RNA gene) [45]. First and second generations of *Terc*-/- *Wrn*-/- mice showed no apparent phenotypic alterations or shortened lifespan. On the other hand, the fourth to sixth generations of the mice with progressively shortened telomeres showed weight loss, reduced survival, premature aging typical of WS, alopecia, cataracts, and severe hypogonadism [45]. These results show the critical importance of WRN helicase in telomere maintenance in mice with shortened telomeres.

It has been reported that the G-rich sequences that form G-quadruplexe structures are scattered in genomic regions other than telomeres at chromosome ends, and that they are involved in the regulation of gene expression [46]. Recent studies have shown that WRN helicase plays a role in bone development and growth by targeting G-rich sequences in the promoter region of the *SHOX* gene [47], which is associated with in the proliferation and osteogenesis of chondrocytes, osteocytes, and their progenitor cells [48], and by promoting transcription of the *SHOX* gene by unwinding the G-quadruplexe structure of the promoter region [47]. This finding clearly implicates defective *SHOX* gene expression in the phenotype which causes short stature in WS patients, and suggests WRN helicase may be involved in promoting the expression of various genes by targeting G-quadruplexe structures in the genome.

## 5. Role in the Regulation of Heterochromatin

It is known that both the epigenomic state, represented by extensive methylation of genomic DNA, and the chromatin structure are drastically altered during cellular senescence. SAHF (senescence-associated heterochromatic foci) is a chromatin signature in senescent cells, and it is the mechanism by which proliferation-related gene expression by heterochromatin, induced by the tumor suppressor gene product, Rb protein, is suppressed [49]. In contrast, suppressive heterochromatin in the α-satellite of the centromere and the satellite II region of the pericentromere are known to decondense and become open regions during aging [50,51].

Using mesenchymal stem cells (MSCs) differentiated from genome-edited human ES cells, Zhang et al. first confirmed that the passaging of cultured *WRN*-deficient MSCs induced premature senescence. They found a decrease in the heterochromatin markers H3K9me3, LAP2β, and LBR in the prematurely senescent WS MSCs, which localize to heterochromatin regions around the nuclear membrane [52]. They also identified a loss of heterochromatin regions in the pericentromere and subtelomere regions in the same cells. Because WRN helicase interacts with H3K9, trimethyltransferase SUV39H1, HP1α associated with trimethylated Lys9 of H3, and LAP2β, and because it contributes to the stabilization of heterochromatin structures in the centromere and pericentromere regions around the nuclear membrane, defective in WRN helicase leads to loss of heterochromatinization and destabilization of these regions (e.g., recombination between repeats, transcription complexes, and replication fork encounters), along with the transcriptional activation of α satellites and satellite II [52].

There are approximately 500,000 copies of the retrotransposons called LINE-1 (L1) on the human genome, accounting for 17% of the genome. Of these, 100–150 copies are functional, encoding single-stranded endonucleases and enzymes with reverse transcriptase activity and moving across the genome [53,54]. In contrast, the host cells recognize retrotransposons in the genome and defend the genome by heterochromatinizing the region very early in the developmental process [55,56,57]. Histone methyltransferases, such as SUV39H and SETDB1, are involved in histone H3 lysine 9 (H3K9) methylation and the formation of heterochromatin by heterochromatin protein 1 (HP1), and thus play an essential role in the heterochromatinization of retrotransposons [58,59].

Extensive epigenetic changes associated with senescence and aging cause retrotransposon reactivation. Decreased expression of RB1 upon cellular senescence initiates the heterochromatinization of the L1 element, and increased expression of FOXA1 promotes L1 transcription [60]. In mice, sirtuin family members SIRT6 and SIRT7 contribute to the repression of L1 expression, and aging-induced depletion of SIRT6 and dysfunction of SIRT7 in the nuclear lamina due to loss of lamin A/C can cancel the transcriptional repression of L1 [61,62]. This massive L1 mRNA accumulation in the cytoplasm during aging may increase cDNA elements in the cytoplasm by reverse transcriptase activity [63]. L1 cDNA accumulation in the cytoplasm leads to cGAS-STING-dependent type I IFN responses and an inflammatory phenotype [60]. In SIRT6 knockout mouse fibroblasts, L1 activation and L1 cDNA accumulation in the cytoplasm cause an increased expression of type I IFN genes [61]. These facts indicate that cytoplasmic L1 cDNA accumulation in the senescent cells maintains the inflammatory phenotype, and premature retrotransposon reactivation may trigger the inflammatory phenotype in the cells with a loss of heterochromatin due to and absence of WRN.

## 6. WRN Helicase as a Molecular Target for Cancer

WRN contributes to chromosome stability not only in normal cells but also in cancer cells. Thus, WRN is a feasible molecular target for anticancer drugs. We have long pursued the possibility of WRN being a potential anticancer target because we have speculated that some cancer cells depend on WRN for their survival [64]. Recent genome-wide screening studies using CRISPR-Cas9 sgRNA libraries have reported that WRN inhibition induces synthetic lethality in cancer cells with high microsatellite instability (MSI) [65,66]. In the cancer cells deficient in mismatch repair, microsatellite sequences consisting of a few nucleotide repeats are unstable, and replication slippage causes large-scale elongation of the repeats to form a unique secondary structure that halts the progression of replication forks. A restart of DNA synthesis from this stalled replication fork requires unwinding of the secondary structure by the WRN, which depends on the activation of the ATR checkpoint kinase. However, when WRN is inhibited in the MSI cancer cells, the secondary structure of the elongated repeats is cleaved by the MUS81 nuclease, leading to massive chromosome disruption [67]. After the replication fork is processed by MUS81 nuclease, the normal cells with complete DNA repair mechanisms resume replication, utilizing several repair mechanisms. On the other hand, in cancer cells, the backup of DNA repair mechanisms and cell cycle control mechanisms, including tumor suppressor genes, are defective, which leads to apoptosis with massive chromosome disruption.

In addition to MSI cancer cells, WRN helicase may be a good anti-cancer molecular target for cancer cells harboring deprotected and unstable replication forks. Indeed, a small-molecule compound that inhibits WRN helicase activity, obtained by screening, sensitizes cancer cells to some DNA-damaging agents, including topoisomerase inhibitors, poly(ADP ribose) polymerase (PARP) inhibitors, and DNA cross-linking agents [68,69], as well as BRCA2-mutated cells [70]. These findings suggest that WRN helicase may function as a guardian of the genome in cancer cells in which certain DNA repair pathways are vulnerable [71], and that WRN helicase inhibitors, such as PARP inhibitors, which act to increase genomic instability, may play an important role in the clinical development of cancer drugs [72].

## 7. Pathophysiological Research Using Pluripotent Stem Cells

If the induced pluripotent stem cells (iPS cells) from WS patients were available, we would be able to utilize them to elucidate the pathogenesis of WS and further expand our scope of research into regenerative medicine and drug discovery. WRN is essential in the synthesis of nascent lagging telomere chains. That is, the 5′-(TTAGGGG)n-3′ repeat. The repeat serves as a template for the lagging telomere synthesis, forms a G-quadruplex, and inhibits the synthesis of lagging telomere strands by replicative DNA polymerase in vitro [73]. As discussed, WRN unwinds the quadruplex structure formed during telomere replication and supports the synthesis of lagging telomere strands [38,39]. When the replication of the nascent lagging telomere strands is stalled in patient cells due to the formation of a quadruplex structure in the G-rich templates, MUS81 or DNA2 nuclease may act on the site to stimulate the resumption of replication [74,75], resulting in the shortening of the lagging telomere strand (Figure 1d). This also leads to premature shortening and dysfunction of telomeres and causes premature senescence [29,38]. Once established, the iPS cells from the patient circumvent premature senescence caused by telomere dysfunction [4,5].

When the telomerase gene *hTERT* is overexpressed, it suppresses premature senescence and immortalizes patient cells, despite the patient cells lacking the *WRN* gene [30]. During the reprogramming process to establish iPS cells, pluripotent genes such as Oct3/4, Sox2, and other pluripotency genes induce the endogenous hTERT gene and immortalize the iPS cells [5]. We therefore presumed that by reprogramming we would induce telomerase activity in the WS cells and avoid premature senescence, so we attempted to establish the iPS cells by introducing the Yamanaka 4 factors into the WS cells. As expected, we were able to establish the WS iPS cells with no differences in self-renewal capacity, pluripotency, ES cell-like morphology, or gene expression profiles compared with the model iPS cells [4]. In these iPS cell lines, the expression of senescence-associated genes, *CDKN1A* and *CDKN2A*, and a SASP factor, IL-6, observed in the parent fibroblasts, were markedly suppressed, and immortalization of the cells was confirmed by long-term passaging culture [4]. However, the differentiated WS iPS cells, by attenuating *hTERT* gene expression, reproduced the premature senescence phenotype of WS, suggesting that premature senescence at the cellular level in WS is due to the dysfunction of a wide range of the chromosome-stabilizing functions of WRN, especially at telomeres [4].

Cheung et al. also demonstrated that telomere dysfunction was suppressed in the WS iPS cells established by the reprogramming of patient fibroblasts cells [76]. The mesenchymal stem cells (MSCs) differentiated from the patient-derived iPS cells were negative for telomerase activity and showed a phenotype of premature telomere shortening with a defect in the synthesis of nascent telomere lagging strands and premature senescence [76]. On the other hand, the neural stem/progenitor cells (NPCs) from the patient-derived iPS cells were positive for telomerase activity and showed no evidence of premature telomere shortening or premature senescence [76]. Furthermore, DNA damage responses were explicitly observed in patients’ NPCs upon telomerase inhibitor treatment, indicating that telomerase avoids premature telomere dysfunction by functionally compensating for the WRN helicase deficiency in particular cell lineages [76]. These results suggest that the premature aging associated with Werner’s syndrome is caused by premature aging in the tissues composed of telomerase-negative stem/progenitor cell lineages such as MSCs (Table 1). In fact, no abnormalities in the brain were reported in studies of Japanese WS patients [77], and no early neurodegeneration associated with dementia has been reported in the central nervous system of patients with WS [78].

As another example, keratinocytes, which comprise the epidermal tissue of the skin, have been reported to evade premature telomere dysfunction caused by WRN helicase deficiency [79]. Unlike fibroblasts in the dermis, keratinocytes possess telomerase activity and thereby play an important role in skin regeneration [80]. In vitro, keratinocytes derived from WS patients showed telomerase activity, and, like those from healthy individuals, they had a mitotic capacity of more than 100 population doubling levels [79]. Therefore, we can conclude that the epidermis is a tissue that may not exhibit a notable phenotype of premature aging due to *WRN* deficiency (Table 1). This also suggests that Werner’s syndrome is a segmental premature aging syndrome.

Taken together, the above suggests that ectodermal-derived or epithelial tissues maintained by telomerase-positive stem/progenitor cells may not contribute significantly to the WS phenotype. Patients with WS have an almost 100% chance of developing cataracts in both eyes [77]. However, its pathogenic mechanism remains unclear. Cataracts are the leading cause of blindness worldwide, and they are attributed to the degeneration of lens proteins [81]. The mammalian eye is formed by interactions between the neuroectoderm, the surface ectoderm, and the periocular mesenchyme [82]. The lens is derived from the surface ectoderm, and the lens placode, the source of the lens, is derived from the surface ectoderm through contact with the ocular vesicle, where the master regulatory gene, *PAX6*, plays a crucial role in the induction and maintenance of the lens placode [82]. The lens is maintained homeostatically by the lens epithelium, in which cells with infinite proliferative capacity migrate from the germinative zone in the equatorial region to the central zone, differentiate into the lens fiber cells, denucleate, and ultimately differentiate into lens fibers [83]. It has been reported that lens epithelial cells in the germinative zone possess telomerase activity in dogs, cats, and mice, but this has not yet been elucidated in humans [83].

Lin et al. investigated cell populations that contribute to the self-renewal of human lens epithelial cells (LECs) using cultured fetal human LECs [84]. The human fetal LECs contained a population of cells that were PAX6, SOX2, and Ki67 positive and expressed BMI-1, which promotes tissue stem cell maintenance and self-renewal. Furthermore, knockdown of BMI-1 markedly reduced LEC proliferative capacity [84]. A report that BMI-1 induces telomerase activity in cells with epithelial lineage [85], and the fact that LECs contain cells with infinite proliferative capacity, together with the results of Lin et al., suggest that mitotically competent BMI-1 positive LECs as well as PAX6 and SOX2 positive human LECs may exhibit telomerase activity. Given these facts and speculations, are LECs in fact involved in developing cataracts in patients with Werner’s syndrome? The length of telomere in LECs in old rats is shorter than that in young rats, suggesting a relationship between the attenuated proliferative capacity of LECs, telomere shortening, and the appearance of cataracts [86]. Thus, age-related telomere shortening in human LECs may be involved in the attenuation of LEC proliferative capacity, and *WRN* deficiency may accelerate senescence in human LECs (Table 1). Alternatively, after LECs undergo differentiation into lens fiber cells and subsequently differentiate into lens fibers, a mechanism different from premature telomere dysfunction involved in mitotic cell division (e.g., epigenetic dysregulation of gene expression involving WRN helicase) may result in pathological changes in the quality and quantity of the proteins that compose the lens [87]. It would be interesting to induce the differentiation of ocular tissues in WS iPS cells to determine whether the pathological state in the lens is due to premature senescence in the LECs or whether it occurs in the terminally differentiated lens fibers. Similarly, since patient-derived epidermal keratinocytes (epidermal stem cells) do not undergo premature senescence [79], it would be interesting to induce differentiation of WS iPS cells into epidermal stem cells to determine whether they have the same mitotic potential as in healthy individuals or whether they undergo any premature senescence at the differentiation stage of the keratinocyte in the stratification of the epidermis.

As mentioned above, recent studies suggest which of the patient’s symptoms are related to molecular-level abnormalities caused by a deficiency of WRN helicase, and the cell types affected as a result (Figure 2). These findings suggest that cataracts are associated with dysfunction in transcription and telomere dysfunction in lens epithelial stem cells [84,87], skin ulcers with telomere dysfunction in mesenchymal cells [27,52,76], short stature and osteoporosis with dysfunction in transcription in mesenchymal stem cells [47], and cancer with dysfunction in repair, telomere dysfunction, and chromosomal instability in mesenchymal stem cells [2,29,76,77]. Based on these relationships between cell types and WRN defects, we can speculate that alopecia is associated with telomere dysfunction in hair follicle stem cells, type 2 diabetes with telomere dysfunction in mesenchymal stem cells, and arteriosclerosis with telomere dysfunction in vascular endothelial cells.

## 8. Human RecQ Homologs, RecQL4 and RecQL5

There are five RecQ homologs in the mammalian genome. *RecQL1* is the *E. coli RecQ* ortholog, *RecQL2* and *RecQL3* are the *BLM* and *WRN* genes responsible for Bloom and Werner syndromes, respectively [2]. We cloned *RecQL4* and *RecQL5* based on the expressed sequence tag (EST) sequences homologous to these three human RecQ-like genes [8,9] and identified a mutation in the *RecQL4* gene in a Rothmund-Thomson syndrome patient [10]. Recently, the *RecQL1* gene was identified as the causative gene for RECON syndrome, a chromosomal instability disorder [7]. All four diseases are manifestations of accelerated aging. Together with various DNA repair and recombination factors, the RecQ-like proteins encoded by the three causative genes of the diseases function in a wide range of DNA repair pathways, including NHEJ, HR, MMEJ, and SSA [2]. These studies have revealed that genetic disruption of mechanisms that maintain the integrity of genomic information induces chromosomal instability, leading to premature aging from cellular level to the level of the whole organism. It has also been suggested recently that RecQ-like proteins are involved in normal aging mechanisms in healthy elderly individuals [88].

## 9. Closing Remarks

As discussed above as an example, we already have numerous data on the interactions and pathways of DNA repair factors from prokaryotes to advanced eukaryotic organisms, including humans. Using neural networks, machine learning, and artificial intelligence technology, we will potentially be able to predict unknown functions of particular repair factors and the pathways that can compensate for the deficiency of already identified and characterized factors. Combined with a single-cell omics analysis of all the cell types that make up a body, it may even be possible to identify diseases caused by mutations in the preidentified factors, their treatments, and synthetic lethal genes in cancer.

As discussed throughout this manuscript, WRN helicase may regulate the DNA repair pathway choice, i.e., the compensatory mechanisms that fill the gap to repair DNA lesions in the patient’s WRN helicase-defecient cells (Figure 3). Consequently, the cells that no longer maintain genomic integrity are likely to exhibit an early cellular senescence phenotype. With current technology, relying on drug discovery to fill this gap is not realistic, because WRN is a nucleic enzyme. Radical treatments are yet to be sought; gene therapy using the *WRN* gene or the *hTERT* gene, or the development of compounds that induce endogenous *hTERT* expression should be pursued. In the future, regenerative medicine might also be an option. Mutant *WRN* genes in a patient’s iPS cells have been corrected by genome editing, and the cells have been differentiated into various types of cells for transplantation. In addition, it has been shown in experiments using mice models that systemic aging is caused by chronic systemic inflammation induced by SASP factors secreted by senescent cells in the body [89]. Drugs that selectively kill senescent cells in the body, called senolytics, are being developed [90], and these may be considered for treating premature aging disorders in the near future.

## Figures and Tables

**Figure 1 genes-13-01802-f001:**
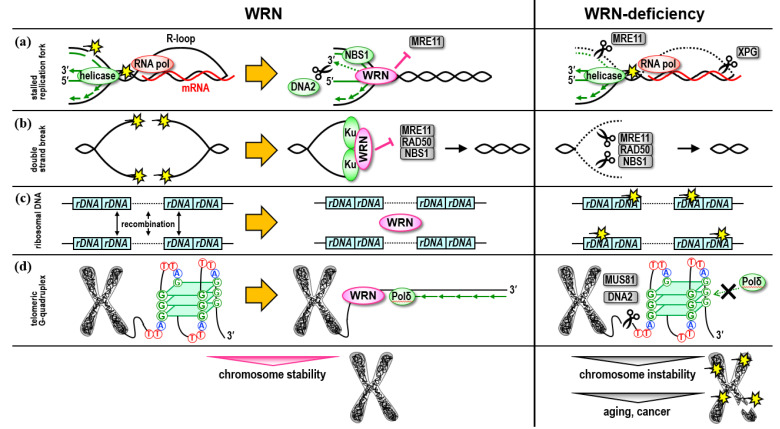
Roles of WRN helicase in DNA repair pathways. WRN helicase is involved in the maintenance of chromosome stability through actions in DNA replication, repair, and recombination via interaction with DNA repair factors, telomere-binding proteins, and other DNA metabolic factors. (**a**) At the reversal site of the nascent DNA strand generated by the arrested replication fork, WRN is thought to support replication fork reassembly. At the reversal site of collision of Pol II complexes with replication forks, the cells preferentially utilize the WRN function to restore transcription and DNA replication. If the WRN does not function, small-scale DNA strand degradation occurs around the replication fork caused by other DNA repair pathways, resulting in the deletion of genetic information and abnormal chromosome structure. (**b**) In DNA double-strand break repair, WRN binds to the DNA ends of the breaks together with the Ku dimer, inhibits the recruitment of the MRN complex, promotes non-homologous end-joining repair, and prevents repair involving deletion of genetic information due to end resection. (**c**) WRN is involved in maintaining a highly regular and uniform rDNA repeat structure in the human rDNA array, suppressing illegitimate DNA recombination. (**d**) WRN helicase supports the synthesis of the nascent lagging strand by replicative DNA polymerase while resolving the quadruplex structure. In WS patient cells, when the replication of the nascent lagging telomere strand is stalled due to the formation of a quadruplex structure in the G-rich templates, MUS81 or DNA2 nuclease may act on the site to stimulate the resumption of replication, resulting in the shortening of the lagging telomere strand.

**Figure 2 genes-13-01802-f002:**
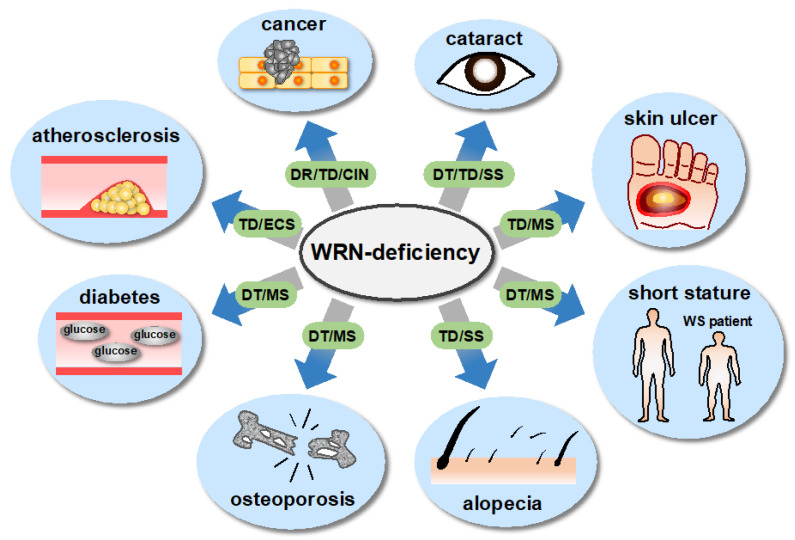
The link between molecular processes associated with WRN helicase and pathological/clinical features. WS patients present with a variety of symptoms associated with aging. Recent studies suggest which of the patient’s symptoms are related to specific molecular-level abnormalities caused by a deficiency of WRN helicase, and the cell types affected as a result. The patient’s symptoms are shown in bold letters in the outermost circle, and the suggested molecular abnormalities and the resulting affected cell types are shown as abbreviations in the middle circle. Note: CIN = chromosomal instability; DR = dysfunction in repair; DT = dysfunction in transcription; ECS = endothelial cell senescence; MS = mesenchymal cell senescence; SS = stem cell senescence; TD = telomere dysfunction.

**Figure 3 genes-13-01802-f003:**
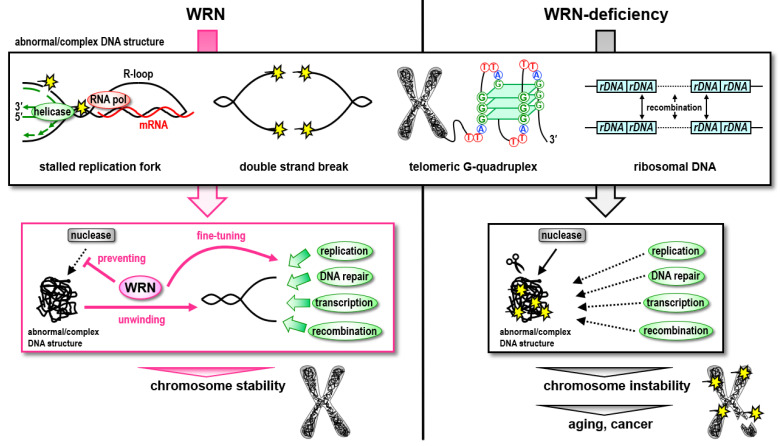
Summary of roles of WRN helicase in DNA repair. WRN helicase may regulate DNA repair pathway choice. The compensatory mechanisms fill the gap to repair DNA lesions in the cells deficient in WRN helicase.

**Table 1 genes-13-01802-t001:** Is the premature aging phenotype associated with the premature senescence of cells of mesenchymal, epithelial, or neural origin?

Tissue	Cell Types	Growth Potential	Telomerase	*WRN*	Premature Senescence	References
Mesenchyme	Fibroblast	+	−	−	+	[27]
	MSC	+	−	−	+	[52,76]
Epithelium	Differentiated keratinocyte	+/−	−	−	?	-
	Epidermal stem cell	+	+	−	−	[79]
Neural tissue	Neuron	−	−	−	−	[78]
	Neural stem/precursor cell	+	+	−	−	[76]

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
