# Peer review of "Research on Werner Syndrome: Trends from Past to Present and Future Prospects"

_genes, 2022, doi:10.3390/genes13101802_

Round 1

Reviewer 1 Report

The following issues should be addressed in a revised version:

Abstract and elsewhere:

WRN is referred to as a gene that encodes a DNA helicase; however, the N-terminal region of the protein contains a conserved 3’ to 5’ exonuclease domain characteristic of proofreading exonucleases. This concept is only considered in Section 4 and should be addressed from the outset.

WRN, the gene, should be italicized. This is not done throughout the manuscript, beginning with the abstract.

Section 1, Introduction:

25 years ago should be 26 years ago—the WRN gene was identified and cloned in 1996

The last paragraph of Introduction only mentions linkage of RECQL4 mutation to Rothmund-Thomson syndrome. This is odd, given the published work demonstrating mutations in BLM and RECQL1 being linked to Bloom's syndrome and RECON syndrome, respectively. While Blom syndrome is mentioned finally in Section 8, the information belongs in Introduction along with other RECQ helicase diseases.

Section 2:

The Section heading should be changed to Roles in replication stress response and DNA repair. This is because a significant portion of this section discusses evidence that WRN is involved in stability of replication forks; this role is distinct from that of WRN’s involvement in double-strand break repair (such as NHEJ, discussed later in the section), for example.

Section 3:

A proposed role of WRN in ribosomal DNA metabolism is suggested. Yet no connection of this potential ribosomal role of WRN in cellular senescence or aging is even mentioned or suggested.

Section 4:

Holiday should be Holliday

CRABBE should be Crabbe

Section 6:

WRN helicase as a molecular target for cancer should include brief discussion of small molecule inhibitors of WRN helicase activity applied in experimental setting to curtail cancer cell growth and proliferation.

Section 7:

Werner syndrome is written out rather than abbreviated WS. At this point in manuscript, WS should be used, as denoted in abstract. Later in the same section Werner’s syndrome is written out rather than abbreviation, as well. Please check entire manuscript for this recurring issue.

Missing papers of significance that are omitted from the manuscript and should be included:

WRN promotes bone development and growth by unwinding SHOX-G-quadruplexes via its helicase activity in Werner Syndrome.

Tian Y, Wang W, Lautrup S, Zhao H, Li X, Law PWN, Dinh ND, Fang EF, Cheung HH, Chan WY.Nat Commun. 2022 Sep 16;13(1):5456. doi: 10.1038/s41467-022-33012-6.PMID: 36114168 

This paper deals with a fundamental advance in our understanding o WRN’s catalytic action that is relevant for bone development and growth.

WRN helicase safeguards deprotected replication forks in BRCA2-mutated cancer cells

Arindam Datta 1Kajal Biswas 2Joshua A Sommers 1Haley Thompson 1Sanket Awate 1Claudia M Nicolae 3Tanay Thakar 3George-Lucian Moldovan 3Robert H Shoemaker 4Shyam K Sharan 2Robert M Brosh Jr 5

Nat Commun2021 Nov 12;12(1):6561.

 doi: 10.1038/s41467-021-26811-w. PMID: 34772932

This paper deals with an important role of WRN in fork protection in cancer as defined by genetic background.

Author Response

GENERAL RESPONSE TO THE EDITORS:

We are pleased, Dr. Jingyun Yang, Dr. Chuntao Zhao, Ms. Clover Wang, and Ms. Foy Qi, that you proposed an invitation for us to revise our manuscript for the Special Issue “Genetics of Complex Human Disease” in the journal Genes. Also, we deeply appreciate you entrusting our manuscript to appropriate reviewers and giving helpful comments. Their recommendations were a huge help to us.

  1. Please revise your manuscript according to the referees’ comments and upload the revised file within 5 days.

RESPONSE: Following the editor's instructions, we revised the manuscript according to the referee's comments and uploaded the revised file within five days.

  1. Please use the version of your manuscript found at the above link for your revisions.

RESPONSE: Following the editor's instructions, we used the revised manuscript downloaded from the link.

  1. Please check that all references are relevant to the contents of the manuscript.

RESPONSE: Following the editor's instructions, we checked that all references are relevant to the contents of the manuscript. In addition, the references listed in Table 1 were checked and modified appropriately.

  1. Any revisions made to the manuscript should be marked up using the “Track Changes” function if you are using MS Word/LaTeX, such that changes can be easily viewed by the editors and reviewers.

RESPONSE: Following the editor's instructions, we used MS Word's Track Changes function to make the corrections. In addition, we used the comments function to indicate the corrections we made to each reviewer.

  1. Please provide a short cover letter detailing your changes for the editors’ and referees’ approval.

RESPONSE: Following the editor's instructions, we have submitted a cover letter explaining our corrections in detail to the editor's and referee's comments, including the statements noted above.

Thank you once again for your valuable comments and suggestions, and for taking the time and making the effort to help us improve the paper.

RESPONSES TO REVIEWER #1’s COMMENTS:

  1. COMMENT: In “Introduction” section 1, WRN is referred to as a gene that encodes a DNA helicase; however, the N-terminal region of the protein contains a conserved 3’ to 5’ exonuclease domain characteristic of proofreading exonucleases. This concept is only considered in Section 4 and should be addressed from the outset.

RESPONSE: We are appreciative of this comment. According to your comment, “with 3' to 5' exonuclease activity” is added appropriately in the text, Abstract and Introduction. See the revision history and the comments box on the right side of the manuscript for more information.

  1. COMMENT: WRN, the gene, should be italicized. This is not done throughout the manuscript, beginning with the abstract.

RESPONSE: We are appreciative of this comment. The entire manuscript was scrutinized and all “WRN” for the gene was corrected to italics. See the revision history and the comments box on the right side of the manuscript for more information.

  1. COMMENT: 25 years ago should be 26 years ago—the WRN gene was identified and cloned in 1996.

RESPONSE: Thank you for this suggestion. We corrected 25 years ago to 26 years ago. See the revision history and the comments box on the right side of the manuscript for more information.

  1. COMMENT: The last paragraph of Introduction only mentions linkage of RECQL4 mutation to Rothmund-Thomson syndrome. This is odd, given the published work demonstrating mutations in BLM and RECQL1 being linked to Bloom's syndrome and RECON syndrome, respectively. While Blom syndrome is mentioned finally in Section 8, the information belongs in Introduction along with other RECQ helicase diseases.

RESPONSE: Thank you for this suggestion. According to your comment, we described mutations in BLM and RECQL1 that are associated with Bloom's syndrome and Recon syndrome, respectively. See the revision history and the comments box on the right side of the manuscript for more information.

  1. COMMENT: In Section 2, the Section heading should be changed to Roles in replication stress response and DNA repair. This is because a significant portion of this section discusses evidence that WRN is involved in stability of replication forks; this role is distinct from that of WRN’s involvement in double-strand break repair (such as NHEJ, discussed later in the section), for example.

RESPONSE: Thank you for this suggestion. According to your comment, the section heading changed to "Roles in replication stress response and DNA repair". See the revision history and the comments box on the right side of the manuscript for more information.

  1. COMMENT: In Section 3, a proposed role of WRN in ribosomal DNA metabolism is suggested. Yet no connection of this potential ribosomal role of WRN in cellular senescence or aging is even mentioned or suggested.

RESPONSE: Thank you for this suggestion. According to your comment, we have described the nucleolar localization of WRN proteins and their function in the nucleolus. See the revision history and the comments box on the right side of the manuscript for more information.

  1. COMMENT: In Section 4, Holiday should be Holliday, and CRABBE should be Crabbe.

RESPONSE: Thank you for this suggestion. We revised them according to your comments. See the revision history and the comments box on the right side of the manuscript for more information.

  1. COMMENT: In Section 6, WRN helicase as a molecular target for cancer should include brief discussion of small molecule inhibitors of WRN helicase activity applied in experimental setting to curtail cancer cell growth and proliferation.

RESPONSE: We agree with you, and according to your comment, we have described small molecule inhibitors of WRN helicase activity and added appropriate references. See the revision history and the comments box on the right side of the manuscript for more information.

  1. COMMENT: In Section 7, Werner syndrome is written out rather than abbreviated WS. At this point in manuscript, WS should be used, as denoted in abstract. Later in the same section Werner’s syndrome is written out rather than abbreviation, as well. Please check entire manuscript for this recurring issue.

RESPONSE: Thank you for this suggestion. According to your comment, “Werner syndrome” is written out only in the Abstract, and abbreviations “WS” were used from the abstract onward. See the revision history and the comments box on the right side of the manuscript for more information.

  1. COMMENT: Missing papers of significance that are omitted from the manuscript and should be included:

RESPONSE: We agree with you, and according to your comment, we cited papers by Tian et al. and Datta et al. at the end of Sections 4, and 6, respectively.

Thank you once again for your valuable comments and suggestions, and for taking the time and making the effort to help us improve the paper.

Reviewer 2 Report

The abstract clearly illustrates the aims of this article. As for the title, it's not very precise, but it's informative enough to draw its potential reader's attention. This article reviews the recent trends in the field, including the utilization of iPS cells in WS research and development of its treatment.he authors take a detailed look at disease mechanisms and drug discovery using the huge amount of scientific data accumulated to date.

In the introduction of this article is successfully explained why the current research is important. They mention in this part and several times in the text that WRN interacts with a great variety of DNA metabolic proteins and is involved in repair, replication, recombination, transcription, epigenetic regulation  etc. It would be better if they could provide a general figure/scheme for the overall participation of WRN in the molecular processes and pathological/clinical features (premature aging, cancer, apoptosis, scalp hair loss, skin and muscle atrophies, etc.)

As for the references, they are relevant, related to the topic, the correctly cited. There are some new articles that may be useful to the authors in better explaining possible cancer targeting therapy with few sentences (Maity, J. ; Horibata, S.; Zurcher, G.; Lee, J.-M.Targeting of RecQ Helicases as a Novel Therapeutic Strategy for Ovarian Cancer. Cancers 2022, 14, 1219. https://doi.org/10.3390/cancers14051219; Gupta P, Majumdar AG, Patro BS. Enigmatic role of WRN-RECQL helicase in DNA repair and its implications in cancer. J Transl Genet Genom 2022;6:147-56. http://dx.doi.org/10.20517/jtgg.2021.60 ;

Dias MP, Moser SC, Ganesan S, Jonkers J. Understanding and overcoming resistance to PARP inhibitors in cancer therapy. Nat Rev Clin Oncol. 2021 Dec;18(12):773-791. doi: 10.1038/s41571-021-00532-x ;

Guterres AN, Villanueva J. Targeting telomerase for cancer therapy. Oncogene. 2020 Sep;39(36):5811-5824. doi: 10.1038/s41388-020-01405-w;)

If the authors intend to emphasize the power of these data as a tool for therapeutic purposes, it is essential to mention, but not necessarily, data on morbidity, age, and mortality in context of WS.

The article is written very detailed and the figures fully illustrate the information. The reader understands the aims and trends although the large volume of scientific data. Providing readers with possible hypotheses and solutions in the context of scientific facts would also be helpful.

All these refinements may make the article more interesting for readers from multiple backgrounds.

Author Response

GENERAL RESPONSE TO THE EDITORS:

We are pleased, Dr. Jingyun Yang, Dr. Chuntao Zhao, Ms. Clover Wang, and Ms. Foy Qi, that you proposed an invitation for us to revise our manuscript for the Special Issue “Genetics of Complex Human Disease” in the journal Genes. Also, we deeply appreciate you entrusting our manuscript to appropriate reviewers and giving helpful comments. Their recommendations were a huge help to us.

  1. Please revise your manuscript according to the referees’ comments and upload the revised file within 5 days.

RESPONSE: Following the editor's instructions, we revised the manuscript according to the referee's comments and uploaded the revised file within five days.

  1. Please use the version of your manuscript found at the above link for your revisions.

RESPONSE: Following the editor's instructions, we used the revised manuscript downloaded from the link.

  1. Please check that all references are relevant to the contents of the manuscript.

RESPONSE: Following the editor's instructions, we checked that all references are relevant to the contents of the manuscript. In addition, the references listed in Table 1 were checked and modified appropriately.

  1. Any revisions made to the manuscript should be marked up using the “Track Changes” function if you are using MS Word/LaTeX, such that changes can be easily viewed by the editors and reviewers.

RESPONSE: Following the editor's instructions, we used MS Word's Track Changes function to make the corrections. In addition, we used the comments function to indicate the corrections we made to each reviewer.

  1. Please provide a short cover letter detailing your changes for the editors’ and referees’ approval.

RESPONSE: Following the editor's instructions, we have submitted a cover letter explaining our corrections in detail to the editor's and referee's comments, including the statements noted above.

Thank you once again for your valuable comments and suggestions, and for taking the time and making the effort to help us improve the paper.

RESPONSES TO REVIEWER #2’s COMMENTS:

  1. COMMENT: The abstract clearly illustrates the aims of this article. As for the title, it's not very precise, but it's informative enough to draw its potential reader's attention. This article reviews the recent trends in the field, including the utilization of iPS cells in WS research and development of its treatment.he authors take a detailed look at disease mechanisms and drug discovery using the huge amount of scientific data accumulated to date.

In the introduction of this article is successfully explained why the current research is important. They mention in this part and several times in the text that WRN interacts with a great variety of DNA metabolic proteins and is involved in repair, replication, recombination, transcription, epigenetic regulation  etc. It would be better if they could provide a general figure/scheme for the overall participation of WRN in the molecular processes and pathological/clinical features (premature aging, cancer, apoptosis, scalp hair loss, skin and muscle atrophies, etc.)

RESPONSE: We appreciate very much your kind words about our manuscript. We are pleased to hear that you acknowledge the value of our review article. In the following sections, you will find our responses to each of your points and suggestions. Thank you for taking the time and effort to review our manuscript.

According to your comment, we add a figure related to the link between molecular processes associated with WRN helicase and pathological/clinical features, as figure 2 and the text associated with this figure at the end of Section 7. See the revision history and the comments box on the right side of the manuscript for more information.

  1. COMMENT: As for the references, they are relevant, related to the topic, the correctly cited. There are some new articles that may be useful to the authors in better explaining possible cancer targeting therapy with few sentences. If the authors intend to emphasize the power of these data as a tool for therapeutic purposes, it is essential to mention, but not necessarily, data on morbidity, age, and mortality in context of WS.

RESPONSE: We are appreciative of this comment, and according to your comment, we cited papers by Gupta et al. and Dias al. at the end of Section 6. See the revision history and the comments box on the right side of the manuscript for more information.

  1. COMMENT: The article is written very detailed and the figures fully illustrate the information. The reader understands the aims and trends although the large volume of scientific data. Providing readers with possible hypotheses and solutions in the context of scientific facts would also be helpful. All these refinements may make the article more interesting for readers from multiple backgrounds.

RESPONSE: Thank you once again for your valuable comments and suggestions, and for taking the time and making the effort to help us improve the paper.
